Epidermal cell death in frogs with chytridiomycosis

Brannelly Laura A. laura.brannelly@pitt.edu laura.brannelly@my.jcu.edu.au 1
Roberts Alexandra A. 1
Skerratt Lee F. 1 2
Berger Lee 1 2
1 One Health Research Group, College of Public Health, Medical and Veterinary Sciences, James Cook University , Townsville , QLD , Australia
2 Faculty of Veterinary and Agricultural Sciences, University of Melbourne , Melbourne , Victoria , Australia
Esteban María Ángeles
Electronic publication date: 2017 Feb 1
Publication date: 2017
Volume: 5
Electronic Location ID: e2925
Received 2016 Aug 10; Accepted 2016 Dec 19
Copyright: ©2017 Brannelly et al.
Copyright year: 2017
Copyright holder: Brannelly et al.
License: This is an open access article distributed under the terms of the Creative Commons Attribution License, which permits unrestricted use, distribution, reproduction and adaptation in any medium and for any purpose provided that it is properly attributed. For attribution, the original author(s), title, publication source (PeerJ) and either DOI or URL of the article must be cited.
License URL: https://creativecommons.org/licenses/by/4.0/

Keywords: Apoptosis, Caspases, Chytridiomycosis, TUNEL, Wildlife disease

Funding: Australian Research Council FT100100375 LP110200240 DP120100811 Queensland Government Accelerate Fellowship Queensland Department of Environment and Heritage Taronga Conservation Science Initiative The project was funded by the Australian Research Council (grants FT100100375, LP110200240, DP120100811 to LFS and LB), Queensland Government Accelerate Fellowship (to AAR), the Queensland Department of Environment and Heritage, and Taronga Conservation Science Initiative. The funders had no role in study design, data collection and analysis, decision to publish, or preparation of the manuscript.

==============================
Background

Amphibians are declining at an alarming rate, and one of the major causes of decline is the infectious disease chytridiomycosis. Parasitic fungal sporangia occur within epidermal cells causing epidermal disruption, but these changes have not been well characterised. Apoptosis (planned cell death) can be a damaging response to the host but may alternatively be a mechanism of pathogen removal for some intracellular infections.

Methods

In this study we experimentally infected two endangered amphibian species Pseudophryne corroboree and Litoria verreauxii alpina with the causal agent of chytridiomycosis. We quantified cell death in the epidermis through two assays: terminal transferase-mediated dUTP nick end-labelling (TUNEL) and caspase 3/7.

Results

Cell death was positively associated with infection load and morbidity of clinically infected animals. In infected amphibians, TUNEL positive cells were concentrated in epidermal layers, correlating to the localisation of infection within the skin. Caspase activity was stable and low in early infection, where pathogen loads were light but increasing. In animals that recovered from infection, caspase activity gradually returned to normal as the infection cleared. Whereas, in amphibians that did not recover, caspase activity increased dramatically when infection loads peaked.

Discussion

Increased cell death may be a pathology of the fungal parasite, likely contributing to loss of skin homeostatic functions, but it is also possible that apoptosis suppression may be used initially by the pathogen to help establish infection. Further research should explore the specific mechanisms of cell death and more specifically apoptosis regulation during fungal infection.

Introduction

Amphibians globally are experiencing the greatest loss in biodiversity of all vertebrate taxa (Stuart et al., 2004). One of the major causes of decline is disease, specifically a fungal parasite, Batrachochytrium dendrobatidis, Bd (Skerratt et al., 2007), which causes the fatal skin disease chytridiomycosis. So far, solutions for minimising Bd related mortality are lacking and research devoted to improving survival rates within these declining populations will be key to conservation management.

Although chytridiomycosis is a superficial epidermal infection, disruption to skin function causes severe loss of electrolytes leading to cardiac failure (Voyles et al., 2009). Global research into the host-parasite dynamics has shown that resistance varies among species, populations and individuals. After demonstrating that vaccination is unlikely to be effective at substantially reducing mortality (Cashins et al., 2013; McMahon et al., 2014), the focus has been on identifying key immune factors that may then be selected for in breeding programs (Scheele et al., 2014; Skerratt et al., 2016). While various potential immune mechanisms—such as antimicrobial peptides (Woodhams et al., 2006a; Woodhams et al., 2006b), cutaneous bacterial flora (Woodhams et al., 2007), immune cell receptors (Savage & Zamudio, 2011; Bataille et al., 2015) and lymphocyte activity (Fites et al., 2014)—are currently being studied, few studies have explored cell death and apoptosis in the skin as an immune mechanism.

Apoptosis is a controlled process resulting in programmed non-inflammatory cell death via the action of caspase proteases and phagocytosis (Fink & Cookson, 2005). It is vital to tissue growth and differentiation, and is associated with elimination of damaged cells and infectious agents, particularly intracellular pathogens (Kim et al., 1998; Barber, 2001; Nogueira et al., 2009; Lamkanfi & Dixit, 2010; Ashida et al., 2011). Hosts utilise apoptosis of infected cells to block replication of pathogens, resulting in clearance of the infection. However, some pathogens can evade the immune system by hijacking the host’s apoptotic machinery: they produce apoptosis inhibitors in order to aid in replication and survival, while others can induce apoptosis to facilitate dissemination or destroy host immune cells (Weinrauch & Zychlinsky, 1999; Hasnain et al., 2003; Faherty & Maurelli, 2008; Hacker, 2009; Lamkanfi & Dixit, 2010). Therefore, the ability of the host and pathogen to control apoptosis can greatly influence disease severity and clinical outcomes.

Apoptosis, and more broadly general cell death, may be a pathology of Bd infection in amphibians, as characteristic epidermal degenerative changes have been observed in Bd-infected frogs by electron microscopy (Berger et al., 2005; Pasmans et al., 2010). Dissociation of epidermal intracellular junctions triggering detachment-induced apoptosis was observed when skin explants were treated with zoospore supernatants in vitro (Brutyn et al., 2012). Transcriptomic studies have shown apoptosis pathways are upregulated in skin of resistant and susceptible frog species (Ellison et al., 2014). Amphibian splenocytes also undergo apoptosis when treated with Bd sporangial (but not zoospore) supernatants in in vitro assays, associated with increased intrinsic, extrinsic and effector caspase activity in these immune cells (Fites et al., 2013). Despite the evidence to suggest that Bd can induce apoptosis of specific cells in vitro, there are no studies that use direct and quantifiable assays, or that explore apoptosis mechanisms during progress of an infection in vivo. Furthermore, it is not known whether the host can effectively use apoptosis as an immune defence against chytridiomycosis.

The aim of this study was to quantify levels of cell death and apoptosis in amphibian skin during experimental infection of two species threatened by chytridiomycosis, the alpine tree frog, Litoria verreauxii alpina, and the southern corroboree frog, Pseudophryne corroboree. To ensure accurate measurement of the apoptotic effect, two methods of detection were utilised (Galluzzi et al., 2009): caspase 3/7 protein assay, and terminal transferase-mediated dUTP nick end-labeling (TUNEL) in situ assay. The caspase 3/7 assay quantifies the activity of effector caspases activated by both the intrinsic and extrinsic apoptosis pathways, while the TUNEL assay detects DNA fragmentation characteristically caused by cell death such as apoptosis, necrosis and pyroptosis (Kelly et al., 2003). This exploratory project characterised measures of cell death and apoptosis in epidermal cells. We tested the hypothesis that cell death and apoptosis was correlated with infection intensity and host survival in order to explore if cell death is a mechanism of disease resistance or cutaneous pathology. If infection load is positively correlated with cell death, this could reflect either an effective immune response or pathology of disease. If a high level of cell death is a pathology of disease we expect that high cell death to be present on animals that succumb to disease. But if apoptosis was a useful immune response, we predict that animals that cleared infection would have higher rates initially or at the time of Bd reductions.

Materials and Methods

Study organisms

Litoria verreauxii alpina that were excess to a reintroduction trial (a total of 300 from two different populations) were obtained from Taronga Zoo, Sydney, and were part of a larger experiment (Brannelly et al., 2016b; Brannelly et al., 2016c). Litoria v. alpina are a declining anuran endemic to the Australian Alps in New South Wales and Victoria, Australia. The species is highly susceptible to Bd, which is the primary cause of decline (Bataille et al., 2015; Brannelly et al., 2015b; Brannelly et al., 2016a). The animals sexually mature at 2 years and these animals ranged from two to three years old and were captive raised under strict quarantine protocols and had never been exposed to Bd. Animals were housed individually in 300 × 195 × 205 mm terrarium with gravel substrate, at a room temperature of 18−20 °C. They were fed ad libitum three times weekly with juvenile (10 mm) crickets (Acheta domestica) (dusted with amphibian vitamins and gut-loaded). Animals were misted twice daily for 60 s with reverse osmosis water. Temperature and humidity were monitored daily.

Pseudophryne corroboree were excess to a breeding program at the Amphibian Research Centre, Pearcedale, Victoria, Australia (a total of 120 adult animals from five different populations). These animals were 5–8 years old and part of a larger research experiment (Brannelly et al., 2016b; Brannelly et al., 2016c). Pseudophryne corroboree are functionally extinct in the wild, and highly susceptible to Bd (Brannelly et al., 2015a; Brannelly, Skerratt & Berger, 2015). These animals were housed in the same conditions as above but on a paper towel substrate that was changed once a fortnight.

Inoculation

Animals were allowed to acclimate to their new environment for at least seven days. All animals were tested for Bd infection prior to the start of the experiment (see ‘Testing for Bd’ below) and all were found to be negative. Cultures of Bd were harvested from tryptone, gelatin hydrolysate, lactose (TGhL) agar plates after incubation at 23 °C for 5 days. Three millilitres of artificial spring water was poured onto the plates and incubated for 10 min to allow zoospores to be released from zoosporangia to create the inoculum. Inoculum was poured off the plates and zoospores were counted using a hemocytometer and then diluted with artificial spring water to achieve the appropriate concentration as described below Samples sizes for each inoculation found in Table 1. Note that the inoculation strain used for obtaining infected samples for the TUNEL assay in L. v. alpina differs from the others, and was used to obtain higher infection rates

Table 1 Sample sizes for all animals inoculated, and the subset of animals used in each assay.

Species	Assay	Total animals	Subset used	
		Inoculated	Control	Bd+	Control	Bd+ early	Bd− cleared	
Litoria v. alpina	TUNEL	6	7	2	2			
	Caspase	27	8	4	8		11	
Pseudophryne corroboree	TUNEL	13	6	10	5	3		

TUNEL assay

Litoria v. alpina used in the TUNEL assay, were inoculated with a New South Wales strain of Bd (WastePoint-L. v. alpina-2013-LB2, Passage number 1). Animals were inoculated with 5 × 105 zoospores in 10 mL of inoculum and held in inoculation containers for 24 h.

Pseudophryne corroboree were inoculated with a New South Wales strain of Bd (AbercrombieR-L.booroologensis-2009-LB1, Passage number 11). Animals were inoculated with 1 × 106 zoospores by applying 3 mL of inoculum onto the venter. Animals were placed in individual 40 mL containers for 6 h, and then transferred back into their terraria. Control animals were mock-inoculated using uninfected agar plates (see Table 1 for sample sizes).

Caspase assay

Litoria v. alpina used in the caspase 3/7 assay were inoculated following the same protocol as for P. corroboree above. Control animals were mock-inoculated using uninfected agar plates.

Data collection

Each week animals were swabbed for Bd infection (see below), weighed to the nearest 0.01 g with a digital scale, and measured snout to venter (SVL) to the nearest 0.1 mm with dial callipers. Animals were euthanized with an overdose of MS-222 when severe clinical signs of chytridiomycosis were displayed (irregular skin slough, leg redness, inappetence, lethargy, loss of righting reflex) and animals appeared moribund in accordance with animal ethics. Clinical signs of disease were seen by L. v. alpina from days 52–70 and by P. corroboree from days 45–83. The experiment ended on day 90, when all control and remaining exposed animals were euthanized.

TUNEL assay

In order to explore cell death in the epidermis of animals that experienced a light infection, 3 P. corroboree were euthanized on day 21 post inoculation. This group of animals will hereby be called “Early Bd+”.

Caspase assay

For the L. v. alpina in the caspase 3/7 assay, toe clips were removed from each animal at the second phalange and immediately frozen at −80 °C. Toe clips were removed weekly until week 3, and then fortnightly until the end of the experiment.

Testing for Bd

We tested for Bd infection by using qPCR on skin swabs (Boyle et al., 2004). We used a standard protocol that involved 45 strokes per animal with a sterile rayon-tipped swab (MW-113; Medical Wire & Equipment), five on the middle of the venter, five on each side of the venter, five on each thigh, and five on the ventral surface of each hand and foot. The swab was gently rotated during and between strokes to ensure the greatest amount of DNA was gathered on the swab. Genomic DNA was extracted from the swabs using the Prepman Ultra kit and 2 min of bead beating to break apart the fungal cell walls. The extract was analysed using quantitative PCR following Boyle et al. (2004), in singlicate (Kriger, Hero & Ashton, 2006; Skerratt et al., 2011; Brannelly et al., 2015b) with a positive and negative control, and a series of dilution standards (made in house from a local Australian Bd isolate) in order to determine zoospore equivalents (ZE) per sample.

TUNEL assay

Upon euthanasia, a subset of L. v. alpina animals (n = 2 control and n = 2 exposed), and P. corroboree (n = 5 Bd− control, n = 3 Early Bd+ animals with light Bd infection, and n = 10 Late Bd+ animals at morbidity) (Table 1) were dissected for skin samples (one sample each from the dorsum, venter and thigh, although no thigh skin was taken for the P. corroboree Early Bd+ animals because that skin was used in a transcriptomics project). Skin was fixed in 4% phosphate buffered formaldehyde for 2 h, and tissues were transferred to 80% ethanol prior to embedding in paraffin wax for histological preparation. Routine histological techniques were used to prepare the tissues for light microscopy following standard methods (Woods & Ellis, 1994). Tissues were dehydrated in a graded series of ethanol, cleared with xylene, and embedded in paraffin (all three skin samples in one block). Tissues were serially sectioned at 5 µm, affixed to hydrophilic glass slides with four serial histosections per slide. Three slides were made per tissue section from the three different areas of the skin per animal and stained in the following order: the first slide was stained with hematoxylin followed by eosin counterstaining (H&E). The next two slides were processed with TUNEL assay following manufacturer’s instructions (ApopTag® Red In Situ Apoptosis Detection Kit, Merck Millipore), followed with a DAPI counterstain. The first of the slides used in TUNEL were the assay slides (where we assessed number of apoptotic cells) and the second slide was a quality control test (with two histosections used as a positive control and the last two as a negative control). For the TUNEL assay slides, cells were counted using fluorescent microscopy at 200× magnification under DAPI fluorescence, then the same section counted for TUNEL positive staining, indicating apoptotic cells. At least 100 cells per animal were counted per skin section. To reach 100 cells, all cells were counted within a field of view. If a field of view contained less than 100 cells, another field of view was selected and counted entirely. The H&E stained histosections were used to ensure the site of infection were the areas used in the TUNEL assay to count apoptotic cells.

Caspase 3/7 assay

Frozen toe clip samples were extracted in 100 µL Buffer A (25 mM HEPES pH 7, 5 mM MgCl2) with two 3.2 mm stainless steel beads in a 1.5 mL microtube. Samples were lysed by four cycles of 1 min bead beating followed by 3 min on ice. Samples were then centrifuged at 4 °C for 5 min at 12,000 × g. Supernatant was collected and used in the assay. As toe samples were very small, reaction volumes were kept to a minimum. Protein concentration of each toe was determined to standardise the sample sizes for the caspase assay. Concentration was quantified using the Bradford assay, with 10 µL Coomassie Bradford reagent (Pierce) and 10 µL of protein extract mixed and incubated at room temperature for 2 min. The samples and BSA standards were run in duplicate or triplicate in a 384 well plate, and the absorbance was measured at 595 nm (POLARstar Omega; BMG Labtech). Caspase 3/7 assay (Caspase Glo® 3/7; Promega) was performed in triplicate in a 384 well plate with 10 µL Caspase Glo reagent and 10 µL protein extract. After mixing, the reactions were incubated in the dark at room temperature for 30 min, after which luminescence was measured (POLARstar Omega; BMG Labtech) (caspase activity). All samples of animals that had severe clinical signs of chytridiomycosis (n = 4), all control samples (n = 8), and a randomised subset of animals that cleared Bd infection (n = 11) were analysed. Only samples through week 7 post inoculation were run, which is the last sampling point where all animals with severe infection were alive, in order to ensure statistical validity of the Bd+ animals.

Statistical analysis

Infection load

Infection loads, or ZE determined from qPCR, of animals were log base 10 transformed. Clinical infections were defined as showing signs of disease with high Bd loads by qPCR (i.e., greater than 1,000 ZE). An animal was considered cleared of infection if the animal was qPCR negative for Bd for at least three weeks. For the L. v. alpina involved in the caspase trial, we defined three infection statuses: control animals, animals that demonstrated severe clinical signs (Bd+) and animals that cleared Bd infection (Bd cleared) animals. Bd+ animals were compared with Bd cleared animals in infection intensity using linear mixed effects models; where individual was repeated, the response variable was infection intensity, and fixed effects were week, infection status and week*infection status. Overall infection load for each group (L. v. alpina caspase trial: Bd cleared and Bd+; P. corroboree TUNEL assay: Early Bd infection and Late Bd infection) were determined by averaging infection load at date of death for P. corroboree and infection load through all time points for L. v. alpina and determining the effect size using Cohen’s d statistic (Altman, 1991) in which a large effect is when d > 0.8. Cohen’s d statistic was calculated in Microsoft Excel.

TUNEL assay statistics

The proportion of TUNEL positive cells to TUNEL negative cells in infected animals and control animals was compared using Pearson’s Chi-Squared test for association. Each tissue type (dorsal, ventral and thigh skin) was analysed as a separate chi-squared test. Following the association test, the strength of association was determined by an odds ratio analysis for TUNEL positive and negative cells of each tissue type. The odds ratio analysis was performed in Microsoft Excel (Altman, 1991). Sample sizes for L. v. alpina were too small to determine significance (n = 2 Bd+, n = 2 Bd cleared), so trends were noted following the chi-squared and odds ratio tests, while significance was determined in the P. corroboree samples (n = 5 control, n = 3 Early Bd+ and n = 10 Late Bd+).

Caspase assay statistics

Caspase 3/7 activity was calculated as caspase activity over protein concentration per sample, and then log base 10 transformed. Weeks 1, 2, 3, 5 and 7 were analysed for the three infection statuses, control, Bd+ and Bd cleared Caspase activity was assessed using linear mixed effects models; where the response variable was caspase activity, individual was repeated, and week, infection status and week*infection status were fixed effects. If infection status*week was a significant interaction we used one-way ANOVAs and Bonferroni’s post hoc test to determine in which specific weeks the infection status groups differed in caspase activity. To determine the change in caspase activity each week, linear mixed effects models were used with the same parameters as above, except the response variable was change in caspase activity. To determine which weeks change in caspase activity varied between groups, one-way ANOVAs were performed using Bonferroni’s post hoc test. The association between caspase activity and infection intensity was performed using a linear regression in Bd inoculated animals only. In order to determine if week or status had an effect a general linear model (GLM) was performed; where log10(Caspase) was compared with infection status, week and log10(ZE). All analyses were performed using SPSS (v21) unless otherwise stated.

Animal ethics

Animal ethics was approved by James Cook University in applications A1897 and A2171 for L. v. alpina and A1875 for P. corroboree.

Results

Bd Infection

Infection progressed as expected for the TUNEL assay individuals: all P. corroboree animals inoculated with Bd developed clinical disease, except for the three individuals euthanized at day 21 and represented as Early Bd+. Animals with clinical disease (Late Bd+) survived between 22 and 83 days post inoculation (average of 59.6 days) and their average infection load at date of death was a near to 1,000 fold higher infection load (793.3 time increase in ZE) than the infection load of animals with light infection loads (Early Bd+), which were euthanized on day 21 post inoculation (Late Bd+ = 5.19 Log10(ZE) ± 0.34; Early Bd+ = 2.29 Log10(ZE) ± 0.89; d = 3.04). In Bd inoculated L. v alpina animals (n = 6) all developed clinical chytridiomycosis and survived 39–63 days post inoculation (average of 52.7 days) and their average infection load at date of death was 3.57 Log10(ZE) ± 0.75.

Of the 27 L. v. alpina inoculated in the caspase infection experiment, all became infected, but only four developed severe chytridiomycosis. The first animal died on day 58 after inoculation and the last on day 71 (average of 63 days). All animals that did not develop severe chytridiomycosis had cleared infection by week 12 post-inoculation. The factors that influenced infection load were week, infection status and week*status (linear mixed effects model: week, F11 = 5.425, p < 0.01; infection status, F1 = 23.763, p < 0.01; week*status, F9 = 3.071, p < 0.01) (Fig. 1). Litoria v. alpina animals that developed severe chytridiomycosis had, on average of all time points, a near 200× higher infection load than animals that cleared infection (194 time increase in ZE) (Died = 3.09 ± 1.22 Log10(ZE); Cleared = 0.80 ± 1.28 Log10(ZE)) (Cohen’s d statistic = 1.30).

Figure 1 Infection intensity over the course of the experiment in animals that succumbed to chytridiomycosis (Bd+) (n = 4) and those that cleared infection (Cleared) (n = 23) after week 12 for the Litoria verreauxii alpinain the caspase trial.

Infection intensity is log10(ZE), and error bars indicate standard error.

TUNEL assay

There was more cell death in infected animals compared with uninfected animals in both L. v. alpina and P. corroboree. The location of TUNEL positive cells in situ differed in Bd+ and control animals of both species. In control animals, low levels of background TUNEL positive cells were evenly distributed throughout the epidermal and dermal layers of the skin (Fig. 2A), but in the Bd+ animals, the TUNEL positive cells appeared more frequently in the epidermis (Fig. 2B). On microscopic examination of the H&E stained sections, clumped Bd sporangia were observed to be scattered through the ventral and thigh areas of skin, but none were seen in the dorsum of either species (Late Bd+), or in the P. corroboree with light infections (Early Bd+). TUNEL positive cells appeared to be more concentrated at infection foci (Fig. 2C), but were also more widespread over the epidermis and in deeper epidermal layers than Bd.

Figure 2 Terminal transferase-mediated dUTP nick end-labelling (TUNEL) in situ assay of infected and uninfected animals.

(A) Bd− control thigh skin section of Pseudophryne corroboree, and (B) Bd+ thigh skin section of P. corroboreestained by in situTUNEL assay. The blue is DAPI staining indicating nuclei of the cells, and the red is the rhodamine stain, which indicates DNA fragmentation characteristically caused by apoptosis. The yellow arrow indicates the position of the Bdcluster seen in (C). (C) P. corroboree section of thigh skin stained with H&E. The H&E section is serial to (B). There is a cluster of empty Bdsporangia (arrow) and a few dark immature sporangia near the skin surface. For (A–C) the epidermis is at the top of the photos. Comparing (B) and (C) shows that the rhodamine stained epidermal cells are concentrated around and below the cluster of Bd and where skin damage is visible such as micro-vesicle formation between basal epidermal cells. 400× magnification and the scale bar indicates 0.03 mm.

In P. corroboree, all three skin types showed an increase in TUNEL positive cells when infected with Bd in both early and late infection (Fig. 3A). In the thigh skin, Late Bd+ animals had 12.01 (95% CI [4.92–26.30]; Odds Ratio: Z = 5.46, p < 0.01) times more TUNEL positive cells than control animals (Pearson’s chi-squared: χ12=44.30, p < 0.01). In the venter skin, Late Bd+ animals have 22.31 (95% CI [5.25–94.82]) times more TUNEL positive cells than control animals (Odds Ratio: Z = 4.21, p < 0.01) and 2.16 (95% CI [1.15–4.03]) times more TUNEL positive cells than Early Bd+ animals (Odds Ratio: Z = 4.21, p < 0.01). The Early Bd+ animals had 10.33 (95% CI [2.37–45.067]) times more TUNEL positive cells than control animals in the venter skin (Odds Ratio: Z = 3.11, p < 0.01; Pearson’s chi-squared: χ22=33.45, p < 0.01). In the dorsal skin, the Late Bd+ animals had 14.38 (95% CI [3.32–62.24]) times more TUNEL positive cells than control animals (Odds Ratio: Z = 3.57, p < 0.01) and Early Bd+ animals had 19.88 (95% CI [4.67–84.20]) times more TUNEL positive cells than control animals (Odds Ratio: Z = 4.05, p < 0.01; Pearson’s chi-squared: χ22=29.45, p < 0.01) but there was no difference observed in TUNEL positive cells of the dorsum in Early and Late Bd+ animals (Fig. 3A).

Figure 3 The proportion of TUNEL positive (TUNEL+) cells per skin type.

(A) The proportion of TUNEL positive apoptotic cells per skin type in P. corroboree with light infection intensity (Early Bd+, n = 3), animals that succumbed to disease (Late Bd+, n = 9) and control animals (n = 10). Error bars indicate 95% confidence intervals of a proportion and ∗ indicates a significant increase in apoptotic cell proportions where (∗a) indicates a difference between control and Late Bd+, (∗b) indicates a difference between control and Early Bd+, and (∗c) indicates a difference between Early Bd+ and Late Bd+ skin samples. There was no thigh skin sample taken for the Early Bd+ group. (B) The proportion of TUNEL positive apoptotic cells in L. v. alpina for control animals (n = 2) and Bd+ clinically infected animals (n = 2), where each individual is represented as a separate bar.

Due to the small sample sizes for infected and control L. v. alpina (n = 2 for each group), only trends could be determined. But there was a higher proportion of TUNEL positive cells in the venter and the thigh skin with no observable difference in the dorsal skin (Fig. 3B) (Pearson’s chi-squared: Venter skin, χ12=5.38, p = 0.02; Thigh skin, χ12=9.198, p < 0.01, Dorsal skin: χ12=1.694, p = 0.19).

Caspase 3/7 assay

Caspase activity was positively correlated with infection load in inoculated L. v. alpina animals (Pearson’s correlation: R64 = 0.463, p < 0.01; Linear Regression: F1,62 = 16.943, p < 0.01) (Fig. 4). However, there was no overall difference between the animals that cleared infection (Bd cleared) and those that developed severe chytridiomycosis (Bd+) (GLM: F1 = 0.079, p = 0.685), or between weeks (GLM: F4 = 0.226, p = 0.717).

Figure 4 The correlation between infection intensity, Log10(ZE), and caspase 3/7, Log10(Caspase) of inoculated L .v. alpina over the course of the experiment.

The correlation between infection intensity and caspase activity is 0.463, and the trend line has an equation of y = (0.229)x + 0.939. There is no difference between Bd+ animals that succumbed to Bd infection (n = 4) and animals that were inoculated and then cleared infection (n = 23), or between weeks of infection.

There was a difference in total caspase activity over time among the three groups (control, Bd + and Bd cleared). The three groups differed over week and week*disease status (Linear mixed effects model: week, F4 = 11.974, p < 0.01; week*status, F8 = 2.139, p = 0.037). There was no effect of week on the control group (Mixed model: F4 = 2.463, p = 0.069) (Fig. 5A). At week 3, there was 48.36% less log10 caspase activity in the Bd+ compared with the control animals, and 41.63% less activity in the Bd cleared animals compared with the control animals (ANOVA: F2,18 = 5.512, p = 0.014; Bonferroni Post-Hoc: control v Bd+, p = 0.046, d = 1.408; control v Bd cleared, p = 0.028, d = 0.923).

Figure 5 Caspase 3/7 activity through week 7 for each group of L. v. alpina.

Caspase activity is defined as the lumiunecence reading controlled for by protein concentration per sample and then log base 10 transformed. The three experimental groups were defined as: Bd+ that succumbed (Bd+, n = 4), controls (n = 8) and Bd inoculated that cleared infection (Cleared, n = 23). (A) The caspase activity (Log10 transformed) for each group per week. (B) The weekly change in caspase activity (Log10 transformed) for each group. Error bars indicate standard error. ∗a indicates the Bd+ group differed significantly from the control group at that week, ∗b indicates the cleared group differed from the control group, and ∗c indicates that the Bd+ group differed from the Cleared group.

When investigating the change in caspase activity each week over the first seven weeks post inoculation, there was a difference among the three groups, with week and week*disease status as important factors (Mixed Model: week, F3 = 5.764, p < 0.01; week*status, F6 = 3.044, p = 0.01), but there was no effect of week on the control group (Mixed model: F3 = 20.004, p = 0.371) (Fig. 5B). The change in caspase activity between weeks 3 and 5 differed significantly among the three groups, with the Bd+ animals increasing in caspase 3/7 activity 15.35 times the change in control animals, and 2.162 times the change in Bd cleared animals (ANOVA: F2,25 = 10.65, p < 0.01; Bonferroni Post-Hoc: control v Bd+, p < 0.01, d = 2.519; Bd+ v Bd cleared, p < 0.01, d = 1.241).

Discussion

In this study we explored cell death and apoptosis in the epidermis of Bd susceptible species as a potential mechanism of disease resistance or cutaneous pathology of chytridiomycosis. We tested the hypothesis that cell death and apoptosis were correlated with infection intensity and host survival, and further we hypothesized that if apoptosis was a useful immune response, animals that cleared infection would have higher rates initially or at the time of Bd reductions. We found that cell death does indeed increase drastically during clinical chytridiomycosis as demonstrated through the in situ TUNEL assay. Because caspase 3/7 levels were correlated with infection load, this suggests that cell death is pathology of Bd. However, because animals that cleared infection had relatively low caspase levels, and did not differ from the animals that eventually developed chytridiomycosis in early weeks following exposure, this does not support our hypothesis that cell death and apoptosis was a useful immune response to Bd infection.

We found an increase in cell death in the epidermis of infected P. corroboree and L. v. alpina compared to the control animals using in situ TUNEL assay, consistent with the apoptosis suggested previously by microscopy (Berger et al., 2005; Pasmans et al., 2010; Brutyn et al., 2012) TUNEL positive cells were located near the site of infection, and occurred on the ventral surface of the animal (thigh and venter skin); but not on the dorsum in L. v. alpina. However, Pseudophryne corroboree demonstrated an increase in TUNEL positive cells as infection progressed in all skin tissues. The location of TUNEL positive cells within the epidermal layers, and their increase over time, is consistent with infection of Bd being the cause of epidermal cell death.

Our qualitative observation of more TUNEL positive cells in the venter and thigh skin of Bd infected animals is consistent with the pattern of Bd distribution noted in other amphibian species. One study that measured Bd infection over the body in two Australian hylids (Litoria caerulea and Litoria genimaculata), showed the dorsum was uninfected or lightly infected, with higher loads on abdomen and thighs (Berger, Speare & Skerratt, 2005b; North & Alford, 2008). We did not quantify Bd loads at each skin site in our two species, but noted a similar pattern by histology with no sporangia seen on dorsal skin. In P. corroboree it was unexpected that increased cell death occurred in dorsal skin. This site distribution, together with the diffuse staining in infected sites, shows that cell death is not localised to infected and adjacent cells and may be associated with diffusion of pro-apoptotic factors from the host or pathogen.

For effector caspase 3/7 activity in L. v. alpina, Bd exposed animals initially demonstrated stable and low levels of apoptosis early in infection (1–3 weeks) despite detectable Bd loads (Figs. 1 and 5A). This observed lower caspase activity early in infection suggests that Bd may suppress apoptosis in order to establish infection, which is particularly prominent at week 3 after inoculation. If the host is able to overcome infection, the caspase 3/7 levels gradually rise and return to normal. However, in animals that eventually displayed clinical signs of chytridiomycosis there was then a rapid and sustained increase of caspase activity over weeks 3–5 (Fig. 5B), which correlates with the timing of high pathogen burden (Fig. 1). This rapid increase is not observed for animals that clear infection, suggesting that rapidly increasing apoptosis was not beneficial and may be a mechanism of pathogenesis by Bd because it was correlated with an increase in infection load and mortality. It remains possible that increased apoptosis is a failed host response to higher burdens in these individuals.

Such pathology of initial suppression of apoptosis followed by a rapid increase has been witnessed in other pathogens such as Shigella flexneri, a bacterium causing diarrhea in humans. The pathogen uses a dual cell death control strategy by producing cytoprotective factors early in infection to aid in replication, followed by necrotic cell death signals later in the infection to enable transmission and host tissue damage (Carneiro et al., 2009). While the apoptotic effect of Bd on amphibian lymphocytes in vitro is known (Fites et al., 2014), host and pathogen mechanisms for this phenomenon are still unclear. Therefore, further experiments are required to confirm whether Bd can stimulate or suppress apoptosis in epidermal cells. Cell-specific effects are seen in other pathogens, for example, Salmonella enterica induces cell death in macrophages (Fink & Cookson, 2006), but suppresses apoptosis in epithelial cells (Knodler, Finlay & Steele-Mortimer, 2005), showing that a pathogen can behave differently in different cell types.

Interestingly, while there is evidence of a higher proportion of epidermal cell death in infected animals at morbidity through the TUNEL endpoint assay, we observed no difference in caspase 3/7 activities between Bd+ and control animals at week 7. This discrepancy may be explained by the two different measures of cell death Caspase 3/7 is an effector enzyme integral to the caspase pathway, but other enzymes within the caspase pathway may also be effective indicators of apoptosis. For example, caspase 1 is involved in the inflammatory response in vivo. Furthermore caspase 8 (extrinsic pathway) and 9 (intrinsic pathway) are both known to be active in response to Bd in lymphocytes in vitro (Fites et al., 2013), and further work should explore the levels of these caspases in order to separate activation in the intrinsic and extrinsic pathways. Furthermore, the last caspase data point was measured at week 7, which is 1–3 weeks prior to the frogs becoming moribund when TUNEL assays were conducted. Therefore, the caspase activity may have increased even more later in infection. Site specific apoptosis may also affect the caspase 3/7 results, as this time course experiment required sampling of toes rather than body skin.

It must also be noted that while the TUNEL assay is most often used to explore apoptotic cells, it measures DNA damage, which can be caused by other cell death mechanism like necrosis and pyroptosis (Kelly et al., 2003). Therefore, the increase in assay positive cells at morbidity may be caused by non-apoptotic cell death pathways that do not involve caspase enzymes. Non-caspase mediated cell death might explain the pattern of increased positive TUNEL assay in the Early Bd+ P. corroboree, which was not mirrored in the caspase assay of L. v. alpina. Alternatively the two species might exhibit different host-pathogen interactions.

This study investigating the role of apoptosis through in situ TUNEL assay and caspase 3/7 presence in the epidermis demonstrates only the initial stages of exploring apoptosis in vivo. The early signs of apoptosis suppression in exposed animals suggest that suppression of apoptosis may be used initially by the pathogen in order to establish infection. Also, the steep increase in apoptosis and cell death in animals that succumbed to disease could explain the disruption to epidermal ion transport that ultimately causes cardiac arrest in clinical chytridiomycosis, although further work on the causal mechanisms of pathogenesis is needed. While more research is needed to determine how apoptosis influences disease outcomes in hosts able to clear infection, these results suggest that apoptosis can be important in the pathogenesis of Bd.

Supplemental Information

Supplemental Information 1 TUNEL Data

Click here for additional data file.

Supplemental Information 2 TUNEL photos of DAPI and Rhodamine stain

Click here for additional data file.

Supplemental Information 3 Caspase data

Raw data for the caspase 3/7 assay for L v alpina

Click here for additional data file.

Supplemental Information 4 qPCR Data

Click here for additional data file.

We would like to thank D Tegtmeier, C De Jong, J Hawkes, K Fossen, S Percival, M McWilliams, L Bertola, M Stewart, N Harney, and T Knavel for data collection and husbandry assistance, and M. Merces for help with dissections. We thank M McFadden, P Harlow and Taronga Zoo for raising the L. v. alpina, and G Marantelli for raising the P. corroboree. We thank F Pasmans, A Martel for advice on apoptosis assays, C Constantine, A Kladnik and R Webb for assistance with TUNEL assay, and T Emeto and W Weßels for help with protocol and kit for caspase 3/7 assay.

Additional Information and Declarations

Competing Interests

Author Contributions

Animal Ethics

Data Availability

The authors declare there are no competing interests.

Laura A. Brannelly conceived and designed the experiments, performed the experiments, analyzed the data, wrote the paper, prepared figures and/or tables, reviewed drafts of the paper.

Alexandra A. Roberts performed the experiments, wrote the paper, reviewed drafts of the paper.

Lee F. Skerratt and Lee Berger conceived and designed the experiments, contributed reagents/materials/analysis tools, reviewed drafts of the paper.

The following information was supplied relating to ethical approvals (i.e., approving body and any reference numbers):

Animal ethics was approved by James Cook University in applications A1897 and A2171 for L. v. alpina and A1875 for P. corroboree.

The following information was supplied regarding data availability:

The raw data has been supplied as Supplementary File.

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
