# Peer review of "Epidermal cell death in frogs with chytridiomycosis"

_PeerJ, doi:10.7717/peerj.2925_

## Round 0.1 · original submission · Major Revisions

Your manuscript should be considerably revised, Please consider all the suggestions of the 3 reviewers in the revised manuscript.

·

Basic reporting

Most of the manuscript was written clearly. The introduction was particularly well written. The one issue I had with the introduction was the discussion of chloramphenicol. It appears that you attempted to connect the use of chloramphenicol to treat chytridiomycosis with a study showing that chloramphenicol induces apoptosis in human keratinocytes. The study sited saw apoptosis in keratinocytes at concentrations over 10-fold higher than used to treat chytridiomycosis (20-fold higher than the MIC) and other studies investigating the role of chloramphenicol on different cell types have conflicting results about whether it induces apoptosis. I would suggests removing this discussion entirely.

The discussion covered some important concepts but there were a few issues that need to be corrected or omitted.
The first issue was with the statement, "other caspase enzymes may be better at predicting apoptosis." Caspase 8 and 9 are upstream initiator caspases that activate effector caspases 3/7. Tracking caspase 3/7 was a correct approach for tracking apoptosis because canonically apoptosis requires activation of caspase 3/7, and caspases 8 and 9 wouldn't necessarily be activated during apoptosis.
The second issue surrounded the discussion of clearing apoptotic cells. This part of the discussion was very unclear and seems like spit balling. Dead cells in the epidermis probably would just continue to move superficially to eventually be sloughed off. There is also minimal recruitment of inflammatory cells such as macrophages to sites of Bd infection, so there would be limited phagocytes to remove dead cells.
The last paragraph is very far reaching; this part of the discussion should be omitted.

In the results section the manner used to describe the differences in infection load is inconsistent (one appears to be wrong) and doesn't take into account that you are measuring infection on a logarithmic scale so 5.19 log(10) is about 1000-fold higher than 2.29 log(10). Please correct this.

The last point I would like to make (and I am restating it throughout the sections of this review) is that you have not demonstrated that apoptosis is occurring in the epidermis. By your own admission TUNEL doesn't specifically label apoptotic cells, and because there is no data suggesting that it is apoptosis you need to be accurate and broadly label this as cell death instead of apoptosis especially in the title.

Experimental design

Please clarify some sections of the Materials and Methods:

1. Please explain why different animals were inoculated differently (different Bd isolates, different doses, and different methods of challenge) between experiments/species.

2. Please note how many sections were analyzed per individual for TUNEL histology.

3. For the caspase 3/7 assays, were the protein concentrations normalized among the samples used for the Caspase-Glo assay? It might make sense not to normalize protein concentration because infection might increase or decrease the amount of protein irrespective of the amount of active caspase present. Because toes were probably different sizes (along the foot, among individuals, and the amount clipped) a better normalization would have been to use a set amount of extract from the tissue (ex: using 90% of 10 mg of tissue to normalize to 9 mg of tissue collected from another toe).

Validity of the findings

This area is in the greatest need of improvement in the manuscript. My concerns are listed below in the order of importance. I do not believe that this manuscript should be published unless these issues are corrected or addressed.
1. The cell death described in the epithelium correlating with Bd infection throughout this manuscript is referred to repeatedly as apoptosis despite an absence of data indicating apoptosis is occurring. The TUNEL assay, as noted in the discussion, doesn’t distinguish apoptosis from other mechanisms of cell death. The experiments quantifying caspase3/7 activity tested the hypothesis that the cell death observed by histology was apoptosis. Because caspase activity was not different or lower in infected animals than in naïve controls these data actually suggest that the mechanism is not apoptosis. Based on the single experiment showing caspase 3/7 activity, I would not rule out apoptosis. Because there is no evidence demonstrating that apoptosis is increased during chytridiomycosis and a shred of data suggesting that apoptosis is decreased, it is premature and probably incorrect to call the cell death observed apoptosis.

2. In the experiment quantifying cell death with TUNEL in histological section of L. v. alpina, there were only two individuals in the control (naïve) group and two in the treated (infected) group. You can’t say there is a significant effect of chytridiomycosis on apoptosis when the N=2. You can report the trend but you cannot make any statistical statement about the effect of infection in this experiment. Where ever you place this data, indicate in the legend that N=2 for both groups.

3. More effort needs to be place in analysis and reporting of the histology. There is some discussion about Bd-infected cells and cells adjacent to Bd-infected cells not labeling positive for TUNEL but nearby cells do label positively. The single image does not clearly show this and no quantification of how close TUNEL+ cells are to infected cells was done. To provide evidence, the figure showing histology in the manuscript needs to contain a representative H&E and adjacent section with TUNEL from each tissue sampled from infected individuals from each species along with at least both stains from a section of healthy skin of a naïve individual. To aid the reader in correlating Bd-infected cells to TUNEL+ cells a third panel for each set of histological sections should show an overlay of the TUNEL on the H&E. If such an overlay is not possible or makes it more difficult to interpret the image, please indicate where Bd cells are in both the H&E and the TUNEL serial sections.

4. One of the major direction PeerJ gives to publishing in their journal is that you need to submit all of your data. Every image of H&E and TUNEL histology is data which went into the study. You need to have these all these data compressed or combined and included as supplement to the manuscript.

Additional comments

Investigation of cell death in the epidermis of amphibians with chytridiomycosis is essential for understanding the underlying pathogenesis of this very important disease contributing to the immense loss of amphibian biodiversity. We still really do not understand what is happening in the epithelium and more specifically the keratinocytes. Unlocking this knowledge will provide answers to why there is such a poor inflammatory response to chytridiomycosis and how Bd actually disrupts the skin leading eventually to death in susceptible species. I applaud the authors for embarking on such research.

While the authors do confirm cell death is occurring in the epithelium during chytridiomycosis at and near sites of Bd colonization (as suggested by previous in vitro and transcriptional work), the evidence is still very preliminary. The preliminary nature of the data make it incredibly difficult to interpret the data because there are no follow up experiments to validate the original findings. This not only makes it difficult to come away with conclusions; it makes it nearly impossible to wrap the data up in a coherent story. I appreciate that the host species used in these experiments are very precious and not necessarily easily obtained for research, but the group should have proceeded doing work in a less precious amphibian model or looked for alternative ways to follow up on the research. The authors have begun an important line of investigation but need to pursue deeper questions of the role of cell death in pathogenesis/resistance and the mechanisms of cell death in order to build a cohesive, high impact study.

At this stage, the data is too preliminary to make many of the statements and conclusions that the authors describe. For instance, throughout the manuscript the cell death observed is called 'apoptosis' when, as noted in the discussion, TUNEL does not uniquely indicate apoptosis. Activation of caspase 3/7, which are specific to apoptosis, would have suggested that the cell death observed by TUNEL was apoptosis. The data, however, show that levels of caspase 3/7 activity are similar or lower in infected animals compared to naive ones suggesting that the cell death was mediated by a different mechanism. The lack of further experimentation prevents anyone from designating what type of cell death is occurring (may still be apoptosis).

I can see that this manuscript may have ended up in PeerJ because there is no preclusion for level of impact or how preliminary the data is; for this reason I believe that the manuscript does have merit to be published in PeerJ, but needs to be rewritten in a manner that accurately reflects the data shown before it can be published (please see comments on specific points).

It appears that this manuscript is being submitted in its current form because this is no longer being pursued as an active project. If this is the case and there is no active interest in continuing to pursue deeper questions, please revise the manuscript and resubmit. However, I would encourage the authors to find a way to develop the research into the high impact story it can be. The discoveries made by such an investigation will drive the field of chytidiomycosis research forward significantly.

Reviewer 2 ·

Basic reporting

The main focus of this paper is understanding the role of apoptosis during disease progression for the amphibian-chytrid system. This is a valuable contribution to the literature, given our limited knowledge on the effects of chytrid infection on amphibian skin. The language used is structurally correct and concise, but throughout the manuscript, there were areas that needed more explanation. Also- the introduction does not clearly set up the contribution of this paper well. For example, Line 43: “It is unknown if the damage to epidermal ion transport is primarily through physical damage to layers or via a fungal toxin.” And the authors continue explaining the difference between the mechanisms. After reading the manuscript though, this is still an unanswered question. A question that the authors do answer is, "does apoptosis increase over the course of infection?"

The authors adhere to the structure of the journal except for the Abstract.
Abstract
Headings in structured abstracts should be bold and followed by a period. Each heading should begin a new paragraph. For example:
Background. The background section text goes here. Next line for new section.
Methods. The methods section text goes here.
Results. The results section text goes here.
Discussion. The discussion section text goes here.

This can easily be adjusted.

Experimental design

In the Introduction, the authors mention that their aim is to quantify the rates of apoptosis in amphibian skin during experimental infection. The authors should also include hypotheses that they can test. For example, does apoptosis correlate to infection intensity? Does the concentration of apoptosis in epidermal vs. dermal layers differ? These seem to be analyses, figures, and results they present, but are not clearly stated.

The methods are explained well- and it is a great contribution to the literature.

Validity of the findings

My major concern with the data is that in some cases the same size per treatment per species is as low as 2 individuals. It is difficult to make firm conclusions on such small sample sizes.

It was a little confusing going through the statistical analyses and understanding the sample sizes to each treatment by each species. Please see general comments below.

Some of the conclusions were also strongly worded and should be more speculative. Please see general comments below also.

Additional comments

Abstract:
- Please mention what species were used in the experiment

Intro:
- Line 38: Solutions for what are lacking?
- Line 41-53: This is a great paragraph and very interesting, but I don’t see how the methods and results answer any of the questions or missing pieces of information outlined in this paragraph.
o For example, Line 43: “It is unknown if the damage to epidermal ion transport is primarily through physical damage to layers or via a fungal toxin.”
o This is still unknown- even after this analysis.
o A question you do answer though is, does apoptosis increase over the course of infection?
o Consider reframing so it reflects the information you present.
- Line 84: “The aim of this study was to quantify rates of apoptosis in amphibian skin during experimental infection….”
o Why would we think that apoptosis differs during infection?
- What hypothesis is being tested?
o Apoptosis correlates to morbidity?
o Concentration of apoptosis in epidermal vs. dermal layers differs?

Methods:
- Lines 98-115: Please include information like total number of animals received, number of clutches the animals came from, and if the ages used are considered adults.
- Sections 2.2, 2.2.1, and 2.2.2 could all be condensed. It seems like each species was inoculated with the same Bd dose using the two methods- but the Bd dose differed between species.
- I found it a little confusing to follow the sample sizes for each method and species. Consider using a table with the following headings:
Species Method Total sample size Control Early Bd+ Late Bd+

- Line 139-141: Was the Caspase assay only done on L. v. alpina?
- Line 144-145: Please include the instruments used to measure snout-to-vent length and the mass.
- Line 168: What Bd strain standards were used?
- Line 168-169: “After inoculation animals were tested once a week until succumbing to disease.”
o This is a repeat from Line 144: “Each week animals were swabbed for Bd infection….”
o Please remove one.
- Line 178-179: “Routine histological techniques were used to prepare the tissues for light microscopy following standard methods.”
o Please use a citation.
- Line 192-193: “At least 100 cells per animal were counted per skin section.”
o How were the 100 cells picked? Were they random? Did you count all the cells per section and made sure there was at least 100 cells?
- Was the “cleared” group assigned post-hoc? How was “cleared” defined? Was it that animals had to have a 0 ZGE on the last day of the experiment? Or did they have 0 ZGE for three consecutive days before the end of the experiment?
- In the statistical analyses section, please refer to generalized linear mixed effects models or linear mixed effects models, instead of “Mixed Models.” The capitalization should also not be there.
- Line 221: Please state what was used as the response variable in that model.
- Lines 222-226: Why was the overall infection load for each group for each species calculated and compared? I would imagine that they are different, and we would be interested in the variations over time by species.
- Line 228-230: “…using Pearson’s Chi-squared test for association; where the number of apoptotoic cells were compared to non-apoptotic cells for each tissue type...”
o Why was the number of cells compared between apoptotoic and non-apoptotic?
o Aren’t we interested in determining if the proportion of apoptotic cells differed between treatments and tissue types? That is also what is displayed in Figure 3.
- Line 231-233: “Following the association test, the strength of association was determined by odds ratio analysis of the pooled data was performed in…”
o I think there is a word missing somewhere- or two sentences became one.
- Line 234-247: It is a little confusing to me why a generalized linear mixed effects model was used and then a one-way ANOVA on the same data. A generalized linear mixed effects model is used when you have continuous independent data (e.g., time), and the ANOVA is used when you have categorical data. In essence, the authors are analyzing the same data twice using different statistical methods.
- There are two lines in the paragraph spanning Lines 234-247 that I think are the same, please remove one or combine:
o Line 236-237: “Caspase activity was assessed using Mixed Models; …”
o Line 240-241: “To determine the change in caspase activity each week, Mixed Models were used with the same parameters as above.”
- Line 244: Why was a logistic regression used to analyze the relationship between caspase activity (i.e., continuous) and Bd infection intensity (i.e., continuous)?
o Logistic regressions are used when the response variable is binary (0 or 1).
- Is there a statistical test comparing the apoptosis quantification of the two methods (TUNEL assay vs. Caspase 3/7 Assay? I ask because on line 87, it says, “To ensure accurate measurement of the apoptoic effect, two methods of detection were utilized.” It seems of interest to me that you would want to compare the two methods and determine if your conclusions would have been similar or different. Or explain what one methods tells you that the other does not.
o This comparison could likely only be done for individuals that had both methods done at the same sampling time point.
- Line 250-251: Did you have to have an approved zoo IACUC?

Results:
- What was the average time to death for the animals in each species?
- Line 257: How many animals did not develop severe chytridiomycosis?
- Line 259-260: When reporting results from the generalized linear mixed effects model, please use a table and include the coefficient estimates. That provides information on the slope and rate of change.
- Line 271-272: Are there stats or a figure associated with this statement?
- Line 274: In the methods, there is no mention of quantifying the difference in apoptosis between the epidermal vs. dermal layers. Consider adding this to the methods, moving the point mentioned on this line to the discussion, or removing it.
- Line 283-284: Why are there 2 p-values reported for one phrase? I see one corresponds to an odds ratio test and the other to a Pearson’s Chi-Squared test, but I don’t know what the chi-squared test is referring to. Please clarify.

Discussion:
- Line 339-340: “Studies measuring Bd infection over the body in two Australian hylids showed the dorsum was uninfected or lightly infected, with higher loads on abdomen and thighs”
o How does this relate to your study? As a topic sentence, this should give me the most important piece of information from the entire paragraph.
- Line 344: Missing a . after P corroboree
- Line 349-350: This is a repeat of your methods, give me the take home message from the results instead.
- Line 355-356: “This rapid increase is not observed for animals that clear infection, suggesting that rapidly increasing apoptosis is a mechanism of pathogenesis by Bd.”
o I do not see the logic in the conclusion.
o How can you disentangle the mechanism from this data?
• The host immune system could have rapidly increased apoptosis, or the pathogen could have induced it.
• Because as the pathogen increases, the host may mount a more robust immune response.
o Please remove this sentence or clarify how you disentangled the two mechanisms.
- Line 357: I disagree that Bd caused a delayed apoptotic response. Looking at Figure 5 panel B- Most of the estimates overlap with 0 on the y-axis, suggesting no change in apoptosis between weeks. The sample size for Bd+ is also very very low (n = 4), to be able to make such strong conclusions. Also- from the stats reported- there only seems to be a significant difference on week 3.
o The data seems to suggest this type of pattern- but there is no strong evidence for this conclusion.
- Line 365: How is the pathogen Shigella flexneri related to Bd? Is it a fungus? What disease does it cause? And in what animals?
- Line 369-373: Consider moving this to the intro. This is an example of a pathogen inducing apoptosis, and does not really discuss the results- but instead calls for more research.
- Line 376-377: “This discrepancy may be explained by the two different measures of apoptosis,…”
o Do the two methods measure different quantities pertaining to apoptosis?
- Lines 380-394: Why was caspase 3/7 picked instead of any of the other enzymes mentioned? This should be included in the methods.
- Line 400: corroboree should be italized.

Figures:

Figure 1: Why isn’t the infection intensity over the course of the experiment shown for both species?

Figure 2: The first sentence in the figure legend does not correspond to the pictures shown. Consider stating what the pictures are of.

Figure 3: Why are sample sizes so low for L. v. alpina?

Figure 4: Is the x-axis log transformed? If so- please label.

Figure 5: Please interpret what it means to be above or below the 0 y-axis line in panel B.

Reviewer 3 ·

Basic reporting

This study uses assays quantifying DNA damage and caspase enzyme activity to frogs infected with the chytrid fungus, a pathogen that causes often lethal disease chytridiomycosis. To my knowledge no previous studies have used these techniques with this disease, despite its relevance to understanding pathogenesis. The research question is stated, and the study is a coherent body of work reporting application of the two techniques to two species of susceptible frogs experimentally inoculated with zoospores of the fungus. The manuscript includes figures illustrating stained skin sections for TUNEL test and graphs reporting quantitative results of both assays. Overall I like the approach and potential of these techniques to reveal previously underappreciated aspects of the pathogenesis of chytridiomycosis, however the manuscript also contains deficiencies that affects the quality of the submission. My main issues are (1) confusing/imprecise use of terms, for example when naming treatment groups, or using the term "rate of apoptosis"; (2) several variables are not defined, and/or their units of measure are not provided, both in text and figures; and (3) the dataset for TUNEL includes a strong outlier, but this is not mentioned in text or shown in Figure 3. Excluding this outlier may drastically change the overall pattern of proportion of cells with DNA damage. I believe the above issues can be addressed by rewriting parts of the manuscript and revising figures.

I have included more specific remarks below:

l. 46 - MacMahon et al 2014 should probably be cited here as a study suggesting that vaccination might work for some species and/or under some circumstances.
ll. 92-94 – authors should explain specifically how correlating apoptotic rates with host intensity and host survival can help understand whether apoptosis is related to disease resistance or cutaneous pathology. In other words, how measurements will be used to evaluate both proposed mechanisms, and what are the expectations in terms of the correlations with apoptosis rates for each mechanism.
l. 101 – Is it just the subspecies being highly susceptible to Bd, or are other populations of L. verreauxi also highly susceptible? (if so, revise accordingly “This subspecies is highly susceptible to Bd, which…”). Is it necessary to specify subspecies name?
ll. 136, 141: check journal guidelines for Petri plate (I’d capitalize)
ll. 135, 141: what is meant here is by flooding uninfected plates with spring water, similarly to what done with the Bd cultures – is that correct? Perhaps best to describe the procedure under the Inoculum section (which would also avoid repeating sentences)
l. 146: explain what these severe clinical signs are, because the criteria were important in choosing the duration of the experiment.
ll.146-148: Awkward sentence; use of “abolished” is confusing here. I suggest: “when animals displayed severe clinical signs of chytridiomycosis, lost their righting reflex, and appeared moribund, in accordance with…”
l. 162: “five on each thigh, and five each limb” is confusing because thighs are part of the hindlimbs, does this mean each thigh was swabbed twice? Or does “limb” refer to just ventral surface of hands and feet? Please revise
l. 169: but animals were euthanized, and did not die of disease – is this correct? In which case, it should say “once a week until euthanasia” or “until displaying several clinical signs of chytridiomycosis”
ll. 172-174: revise naming of the groups. Why are there two names for control, ie “control” in line 172 and “Bd-control” in line 173, which suggests that there is some difference in the procedure (if not, please be consistent and use same term throughout). If “Bd-control” just means non-inoculated animals, I’d just call them control (adding Bd- might create confusion). Similarly for infected animals, why are these named “exposed” for one species, but infected for the other species.
ll. 212-214: following from above, for the caspase assay there seem to be the additional group of “animals that cleared Bd infection”. At this point I’m very confused about all these different groups, because the inoculation procedure did not seem to differ between the two assay experiments.
l. 217: “infection loads” need to be defined – is this a result from the quantitative PCR?
ll. 220-221: keep syntax consistent to facilitate reading (use ll.237-238 as example)
l. 221: explain what is meant by “status” – is this infected, non-infected?
l. 222: “Bd –“ seems to be yet another name for “control” or “Bd-control”. Please use same term throughout.
l. 224: “all infection loads” ? “at time of euthanasia”
l. 227-229: this sentence should be rewritten. I’m not sure how a proportion can be turned into a rate, and what rate exactly means in this context. I understand “rate” as being a ratio of number of apoptotic cells per unit of time – is this the intended meaning here? (I doubt it since the time information is not available). Ll. 229-230 “where the number of apoptotic cells were compared to non-apoptotic cells” basically means that the ratio was calculated, i.e. the proportion as already indicated on line 227. Thus, did the analysis simply compare proportions of apoptotic cells among tissue types?
ll. 226, 233: is Excel reliable for these types of analyses?
l. 236: some new variation of group names – see above.
l. 245: what is meant by “status” – is this infection status
l. 246: ZE needs to be defined
l. 274: “low levels…were evenly distributed”
l. 277: “than just at the localised“
l. 349: changes over time since infection?
ll. 351-352: unclear from methods how rate of apoptosis was measured. It seems to me that this term is used to refer to changes in caspase activity, and to avoid confusion I’d use the specific term rather than “rate of apoptosis”. What makes this confusing is that “rate” is also used in reference to proportions of apoptotic cells, which cannot be related to rate.
l. 357: “This delayed apoptotic response suggests” or “These delayed apoptotic responses suggest”
ll. 357-358: But the data do not suggest that apoptosis is suppressed, they only show that high apoptosis frequency is associated with increased pathogen burden. I don’t think a correct interpretation of the data is to suggest suppression of apoptosis at low infection loads, although it is certainly a possibility – this could be rephrased. The text should also explain to which rate of apoptosis the “suppression” is being compared to.
ll. 358-359: this sentence seems to contradict findings presented in Figure 4 that shows no difference in caspase activity between Bd cleared and Bd+ frogs.
l. 374: “of greater proportions of apoptotic cells..”
ll. 376-379: could time lag also be a possible explanation for such discrepancy? (I see this is mentioned later lines 385-386)
l. 391: unclear what the two processes are
l. 405: I’d remove “successful”

Figure 3: I compared this figure with the raw data provided and in my opinion the figure does not provide an accurate representation of results. First, sample size for 3A is 2 individuals, and I don’t think it makes much sense to calculate average, error bars and statistically compare them with such reduced sample size. Second, I see from Column F (which should say ‘proportion cells apoptotic’ and not %, since the values are not percentages) that individual ID#21 is clearly an outlier with 34% apoptotic cells, whereas in none of the other individuals such proportion exceeds 5%. Excluding this single outlier changes quite drastically the overall pattern of proportion of apoptotic cells. Thus, if authors believe there are solid grounds for including the outlier, they should provide the rationale for doing so; and/or mention what happens when this outlier is excluded.
Figure 4: what are the units for the x-axis? Are these log10 (zoospore equivalents) as in Fig 1. Similarly, what are the units for the y-axis.
Figure 5: see above, please indicate units for Caspase (Log10) – is this the same variable as Caspase activity?

Experimental design

The submission describe original primary research within the scope of the journal.

The research question is defined, but the use of terms is somewhat confusing and some sentences lack clarity and precision. This might be due to semantic issues (I believe that is the main reason), which would be easy to address, or to more serious flaws. For example, it is difficult to assess the experimental design given the issue with (same?different?) treatment groups receiving different names in the manuscript.

Research conforms to ethical standards and has been approved by the appropriate university committee.

Validity of the findings

Overall the data are interesting and worth publishing, particularly the caspase activity data. I have some issues with Figure 3 (TUNEL assay data) mentioned above because of the presence of one outlier (see above). I would also rephrase some parts of the discussion that seem excessively speculative, specifically concerning apoptosis suppression at the early stages of infection. This is a possibility but I don't think the data collected here suggest apoptosis suppression is necessarily occurring (alternative explanations for this pattern should also be provided).

---

## Round 0.2 · Minor Revisions

Your manuscript still needs to be further improved. Please follow the suggestions of the reviewers.

·

Basic reporting

No further comments; earlier concerns appropriately addressed

Experimental design

No further comments; earlier concerns appropriately addressed

Validity of the findings

My concerns were mostly addressed, but two points were not fully addressed.

1. The TUNEL data shown for L. v. alpina more accurately show the data. However, because there are only 2 frogs/treatment and only 1-2 slides per tissue investigated, there is very little confidence that the data are actually representative. I think it is fair to show the data and note the trend, but it is too much to force statistical tests on the limited data to imply significance. Please edit the figure and corresponding text.

2. You either forgot or completely ignored the request to upload the histology images. This journal is pretty definitive about including all data ("PeerJ is committed to improving scholarly communications and as part of this commitment, all authors are responsible for making materials, code, data and associated protocols available to readers without delay...Any supporting data sets for which there are no suitable repositories may be made available as publishable Supplemental Information files by PeerJ") The histology images are the raw data not the counts which are your analysis/observation of the data. There appear to be 18 tissue sections from L. v. alpina and 49 from P. corroboree (times 3 for serial sections with different staining). According to the guidelines for PeerJ, all 201 images should be included in supplement because these are raw data (probably most easily by including all in a single pdf file). I would be happy to see a representative TUNEL and DAPI image from each bar shown in Fig. 3 (8 different tissue samples from P. corroboree and 12 from L. v. alpina). These added images would help demonstrate the point being made in lines 280-289 and would assist in trends apparent in L.v. alpina (aiding the above point).

Additional comments

Please make the two modifications in the "Validity of Findings" section. I feel it is most important to include more of the histology images; this paper should not be published without including these images either in the text or supplement.

Reviewer 2 ·

Basic reporting

The manuscript has greatly improved since the last time I saw it, but I am still concerned about the framing of the article. For example, in the abstract under the Background and Discussion sections, the authors set up the problem statement as: “Apoptosis (planned cell death) can be a damaging response to the host but may alternatively be a mechanism of pathogen removal for some intracellular infection” and resolve the discussion by saying, “Future research should explore the specific mechanisms of cell death and more specifically apoptosis regulation during fungal infection”. As a reader, this is very unsatisfying.

The authors rebuttled my last set of comments on this issue (previous review:
Line 43: “It is unknown if the damage to epidermal ion transport is primarily through physical damage to layers or via a fungal toxin.” And the authors continue explaining the difference between the mechanisms. After reading the manuscript though, this is still an unanswered question.) by saying that this is all background information, which I agree with, but I still believe that the way the background information is presented is misleading and does not frame an answerable question well.

Line 85-87: “We tested the hypothesis that cell death and apoptosis was correlated with infection intensity and host survival in order to explore if cell death is a mechanisms of disease resistance or cutaneous pathology”
This is a very interesting hypothesis, but I am unclear how correlations between the two sets of variables (cell death vs. infection intensity; cell death vs. host survival) helps delineate between the two mechanisms (disease resistance vs. cutaneous pathology), but consider adding a mock figure that helps understanding how you interpret your results to your conclusions for each scenario. For example, an image where the x-axis is Bd infection intensity and y-axis is caspase activity. Then, draw the relationship between these two variables for the "Cleared" and "Bd+" groups, and explain how you reach different conclusions (i.e., amphibian immune response vs. Bd pathology) based on those patterns.
It seems like this is the relationship that the authors are basing their conclusions on, but I am confused on how it was interpreted. This figure would help clarify.

Line 140: Consider splitting up the Data Collection section into 3 parts so it mimics inoculations.
1. Data collected in both experiments
2. TUNEL assay
3. Caspase assay

Line 172: Why was no thigh skin taken for P. corroboree Early Bd+?

Line 213: Consider splitting up the Statistical analysis section into 3 parts so it mimics inoculations and data collection.
1. Summary stats
2. TUNEL assay stats
3. Caspase assay stats

Line 226: Mention what you consider to be a small, medium, and large effect size.

Line 346: It is still unclear to me how the authors are interpreting the results to suggest that cell death is pathology of Bd and not an amphibian immune response. Please try to make it clearer in the results how you are interpreting this pattern. The mock figure would also help me understanding what it is that the authors were looking for and how they interpreted their results.

Line 375: What is normal? Pre-Bd infection levels of cell death? Please clarify.

Line 402: Why was the last caspase data point measured on week 7 (1-3 weeks prior to the frogs becoming moribund)? I looked in the methods for an explanation but could not find one. Line 152-153 mentions: “Toe clips were removed weekly until week 3, and then fortnightly until the end of the experiment”- but, this only provides an explanation for missing samples from animals that died between the 2-week sampling schedule, not 3 weeks.

Figures: I think the legends on figures 3 and 4 are swapped.

Figure 3B: Combine the 2 bars for each treatment*sample location and add standard error bars.

Fig 3 and 5A: have the same y-axis but labeled differently. Please be consistent.

Experimental design

See basic reporting- authors do not answer the question they set out to answer, or at least in the presentation of the manuscript as is, it is difficult to understand how they made specific conclusions from the discussion.

Validity of the findings

See basic reporting.

Additional comments

Overall, the authors did a good job addressing the majority of my concerns, except for reframing the question or explaining how they reached particular conclusions.

---

## Round 0.3 · Minor Revisions

Thank you for improving your manuscript. However, it still needs some final changes before being accepted for publication:

1. Please, condense the information on lines 159-162.
2. Line 165: change “inoculation .” by “inoculation.”
3. Line 191: please, remove “tissue” skin is an organ.
4. Line 343: An space has to be added before “Dorsal”.

---

## Round 0.4 · accepted · Accept

Thank you for improving your manuscript.